# Prediction and practical application of bauxite mineralization in Wuzhengdao area, Guizhou, China

**Shiqiang Yang[1], Wu Yang◉[2]\*, Tao Cui[2], Min Zhang[2]**

**1** 106 Geological Party, Guizhou Bureau of Geology and Mineral Exploration and Development, ZunYi, China, **2** Faculty of Resources and Environmental Engineering, Guizhou Institute of Technology, Guiyang, China

\* yangwu@git.edu.cn

**Data Availability Statement:** All relevant data are within the manuscript and its Supporting Information files.

**Funding:** This research was supported by Guizhou Science and technology innovation talent team

## Abstract

Wu-Zheng-Dao District in China is the world's most famous mining areas. It hosts several world-class deposits, such as Xinming, Datang and Luolong bauxite deposits. Although this area still has significant potential for the discovery of new deposits, mineral prediction has become increasingly diffcult as the number of shallow deposits diminishes. Therefore, it is necessary to explore new and effective metallogenic prediction methods.Weights of evidence and machine-learning algorithms were used for mineral prospecting in this study. This study used a confusion matrix, receiver operating characteristic (ROC) curve,and prediction efficiency curve to evaluate the prediction results of each machine algorithm. The results showed that 95.9% of the deposits were located in high and distant scenic areas, accounting for 10% of the total area.The prospectivity map of the Wu-Zheng-Dao district shows that the high prospective areas are generally confined to the claystone and carbonatite rocks of the Eastern region, in particular, of the clay layers, and several areas of high prospectivity also occur in the Southern Cross Domain. According to the predicted results, after on-site exploration, design, and construction, Yanfengqian bauxite deposit was discovered, with an average thickness of 1.82 meters; The average content of $Al_2O_3$ is 61.24%; The resource amount is 28.9503 million tons.

## 1. Introduction

Mineral resources are an important component of natural resources and are a crucial foundation for the development of human society. Mineral resource prediction and evaluation are scientific predictions and comprehensive evaluations of the possible locations and resource potential of economically valuable minerals. In the 1950s, a team led by French scholar Allais conducted the first systematic and decision-supportive mineral resource evaluation work in the Sahara Desert region of Algeria, North Africa, opening a new chapter in the quantitative prediction of mineral resources. Machine learning is currently widely used in mineral resource prediction and evaluation systems, and plays an important role in extracting and integrating

project (CXTD[2023]008, the High-level talent introduction program for the Guizhou Institute Of Technology (0203001018040), Special Projects and Research Topics of Guizhou Association for Science and Technology(0201011022001). Special and research project of Guizhou Association for Science and Technology titled "Research on the Exploration Practice and Selection Theory of Lithium Resources in Northern Guizhou under the Target of Carbon Peak and Carbon Neutrality. SY, WY, TC, and MZ are the recipients of the funding awards listed above. The funders had no role in study design, data collection and analysis, decision to publish, or preparation of the manuscript.

**Competing interests:** The authors have declared that no competing interests exist.

predictive element information. Bayesian theorem, least squares method, and Markov chain method are widely used techniques in machine learning today. Therefore, from this perspective, machine learning was developed several centuries ago, and classic methods in mineral prediction, such as the evidence weight method and logical regression, should also belong to the category of machine learning. Since the 20th century, especially since Alan Turing proposed the establishment of the first learning machine in 1950, deep learning has been widely applied in practical applications, such as facial recognition, speech recognition, and speech translation. In the early 21st century, significant progress was made in modern machine learning. Overall, machine learning involves learning a certain pattern or model from data,and using it to solve practical problems. It is effective in handling nonlinear and high-dimensional data,and is widely used in many disciplinary fields. Since the 1970s, when mineral prediction entered the quantitative stage, machine learning and related data-mining methods have been widely introduced and have become an important research direction in mineral prediction.

Wu-Zheng-Dao District in China is the world's most famous mining areas subject to over a century of mineral exploration. It hosts several world-class deposits,such as Xinming,Datang and Luolong. Although this area still has significant potential for the discovery of new deposits, mineral prediction has become increasingly diffcult as the number of shallow deposits diminishes The prediction model based on GIS is one of the effective means for comprehensive metallogenic prediction, because: (1) it can objectively and accurately evaluate the control of various prospecting factors (including magmatic rocks, strata, structures, geophysical and geochemical anomalies, etc.) On metallogenesis in the region; (2) It can effectively and comprehensively study the prospecting factors of different sources in the region, select appropriate spatial mathematical models, summarize the spatial distribution law of various prospecting factors and deposits, and predict and evaluate the metallogenic potential in the region.

Based on the rock geochemical data of different geochemical data in the Wu-Zheng-Dao area, this study selected various prospecting factors, adopted evidence weights and used machine learning to predict mineralization. The parameters of different algorithms and the influence of sample selection on the prediction results are summarized, and the prediction results of different models are comprehensively evaluated.Spatial analysis and evidence weight analysis are used to analyze the correlation between each prospecting factor and the spatial distribution of deposits in Wu-Zheng-Dao area, and the evidence weight method is used to predict mineralization. [1–4].

## 2. Data and methodology

The geochemical data of the study area were obtained from the GSWA geochemical database and the lithogeochemical survey data from the GA Ozchem database, with a total of samples 22900. Due to the uneven spatial distribution of the original geochemical data, some areas are relatively sparse, in order to facilitate the generation of a unified evidence layer, the original data need to be interpolated:

$$\gamma(h) = \frac{1}{2N(h)} \sum_{i=1}^{N(h)} \left[ T(x_i) - T(x_i + h) \right]^2. \qquad (2-1)$$

where represents the number of sample pairs at a distance. To study the relationship between the sample spacing and the sample value, the sample values obtained by different calculations are projected onto the coordinate system with and as the coordinate axes, and the variogram is obtained. The variation function chart can reflect the trend of the difference of sample values with the change in sample spacing: when the spatial positions of two samples coincide, it is zero; as the value increases, the difference between the sample values also increases; when it

increases to a certain extent, the sample values will become irrelevant to each other, so it becomes constant. The Matheron model is a typical variogram. The mathematical expression of the model is:

$$\gamma(h) = \begin{cases} 0, & h = 0 \\ C_0 + C\left[1.5\dfrac{h}{a} - 0.5\left(\dfrac{h}{a}\right)^3\right], & 0 < h < a \\ C_0 + C, & h > a \end{cases} \qquad (2-2)$$

where is the nugget value, which reflects the difference in the sample value when it is extremely small, and represents the randomness of the sample value; is the range, which indicates the distance range within which the sample values have spatial correlation, that is, beyond this distance, the sample values will become independent of each other; is the arch height, which indicates the degree of the difference of the samples on the effective scale with correlation; and is the sill value, which is used to represent the overall spatial variability of the sample value [5–8].

## 2.1. Construction of training sample set

Positive and negative sample sets were constructed according to the established classical model. There are 171 known ore spots in the study area, and a certain number of negative samples are needed when using a machine learning algorithm to predict mineralization;Therefore, 171 non-ore spots were randomly and evenly selected according to the spatial location in the study area, and the sample set with ore was taken as the positive sample. The sample set w,ithout ore was taken as the negative sample and, they constituted a training sample set containing 342 samples.

To compare the prediction accuracy of the different assignment methods, two types of training sample sets were constructed:

1. Discrete training sample set: whether there is abnormal information in each grid sheet of the statistical evidence layer. If so, the grid cell value is 1; Otherwise it is 0.

2. continuously train a sample set, namely, directly assign a sample value or a sample interpolation result to each grid unit of that evidence.

To analyze the influence of the number of samples on machine learning, three different training sample sets were constructed according to 10%, 40%, and 70% of the total number of samples. At the same time, in order to avoid the impact of random selection of samples on the results, the same number of sample sets were extracted three times, and the final sample sets are shown in Table 1.

**Table 1. Training sample set.**

| Sample set | type | positive samples Number | negative samples Number |
|:---:|:---:|:---:|:---:|
| S1-S3 | Discrete | 18 | 18 |
| S4-S6 | Discrete | 69 | 69 |
| S7-S9 | Discrete | 120 | 120 |
| S10-S12 | Continuous | 18 | 18 |
| S13-S15 | Continuous | 69 | 69 |
| S16-S18 | Continuous | 120 | 120 |

## 2.2. Confusion matrix

Accuracy evaluation is an important means of evaluating machine learning algorithms. It compares the p rediction results with the real data in the test set, counts the number of correct classifications to obtain the error matrix, and then measures the efficiency of machine learning algorithms through various indicators [9–14].

A confusion matrix is a commonly used classification accuracy evaluation method. It intuitively reflects the contrast information between the predicted class of the classification system and the real class; therefore it is often used to evaluate the classification effect of the binary classifier. The confusion matrix is a 2*2 matrix that contains four groups of records, as shown in Table 2.

True Positive (TP): the positive records were correctly classified by the classifier, that is, the number of positive samples correctly classified.

True Negative (TN): The negative class record is correctly classified by the classifier, that is, the number of negative samples correctly classified.

False Positive (FP): records of positive classes misclassified by the classifier, that is, the number of positive samples misclassified.

False Negative (FN): The negative class record is misclassified by the classifier, that is, the number of negative samples misclassified.

In Table 2, the number of records on the main diagonal line indicates the number of elements that are correctly processed by the classifier, and the number on the off-diagonal line indicates the number of elements that are incorrectly classified. Therefore, the larger the number on the main diagonal, the better the classifier.

The classification evaluation index based on the mixed matrix mainly includes Classification Accuracy, Classification Error, Sensitivity, and Specificity. False Positive Rate, False Negative Rate. The calculation formulae are as follows:

$$\text{Classification Accuracy} = (TP + TN)/(TP + FP + TN + FN) \qquad (2-3)$$

indicates the proportion of correct classifications of the classifier, that is, the proportion of the number of correctly classified samples to the total number of samples.

$$\text{Classification Error} = (FP + FN)/(TP + FP + TN + FN) \qquad (2-4)$$

indicates the proportion of misclassification of the classifier, that is, the proportion of the number of misclassified samples to the total number of samples.

$$\text{Sensitivity} = (TP)/(TP + FN) \qquad (2-5)$$

Proportion of correctly identified positive samples to the total number of actual positive samples.

$$\text{Specificity} = (TN)/(TN + FP) \qquad (2-6)$$

Proportion of correctly identified negative samples to the total number of actual negative

**Table 2. Confusion matrix.**

|  |  | Forecast category | |
|---|---|---|---|
|  |  | Positive class | Negative class |
| True category | Positive | True Positive | False Negative |
|  | Negative | False Positive | True Negative |

samples.

$$\text{False Positive Rate} = (\text{FP})/(\text{FP} + \text{TN}) \qquad (2-7)$$

Proportion of the number of positive samples identified incorrectly to the total number of actual positive samples.

$$\text{False Negative Rate} = (\text{FN})/(\text{TP} + \text{FN}) \qquad (2-8)$$

The proportion of negative samples was misidentified as the total number of actual negative samples.

Kappa coefficient is also an important classification progress evaluation index based on confusion matrix, and its calculation formula is:

$$K = (P_O - P_C)/(1 - P_C) \qquad (2-9)$$

The value range of the kappa coefficient is [0, 1]. The closer the coefficient is to one, the better the classifier effect, and vice versa, the higher the reliability of the classification result.

## 3. Mineral prediction based on logistic regression

In this study, logistic regression was first used to predict all the training samples, and a confusion matrix was established according to the prediction results, as shown in Table 3.

As shown in Table 3 that the randomness of the training samples has an impact on the prediction results. When the number of samples was small, the fluctuation in the prediction

**Table 3. Prediction accuracy of logistic regression model for each sample set.**

|  | Accuracy | Mis classification rate | Sensitivity | Specificity | False positive rate | False negative rate | Kappa coefficient |
|---|---|---|---|---|---|---|---|
| **S1** | 75.38 | 24.62 | 71.24 | 75.92 | 24.08 | 28.76 | 0.281 |
| **S2** | 64.12 | 35.88 | 85.62 | 61.32 | 38.68 | 14.38 | 0.211 |
| **S3** | 64.05 | 35.95 | 79.08 | 62.08 | 37.92 | 20.92 | 0.189 |
| **Mean value** | 67.85 | 32.15 | 78.65 | 66.44 | 33.56 | 21.35 | 0.227 |
| **S4** | 90.34 | 9.66 | 82.35 | 91.07 | 8.93 | 17.65 | 0.530 |
| **S5** | 91.82 | 8.18 | 80.39 | 92.86 | 7.14 | 19.61 | 0.578 |
| **S6** | 96.07 | 3.93 | 73.53 | 98.12 | 1.88 | 26.47 | 0.736 |
| **Mean value** | 92.74 | 7.26 | 78.76 | 94.02 | 5.98 | 21.24 | 0.61 |
| **S7** | 97.95 | 2.05 | 70.59 | 99.25 | 0.75 | 29.41 | 0.747 |
| **S8** | 98.75 | 1.25 | 72.55 | 100 | 0 | 27.45 | 0.834 |
| **S9** | 98.57 | 1.43 | 68.63 | 100 | 0 | 31.37 | 0.806 |
| **Mean value** | 98.42 | 1.58 | 70.59 | 99.75 | 0.25 | 29.41 | 0.795 |
| **S10** | 66.84 | 33.16 | 85.62 | 64.39 | 35.61 | 14.38 | 0.235 |
| **S11** | 81.42 | 18.58 | 78.43 | 81.81 | 18.19 | 21.57 | 0.398 |
| **S12** | 63.97 | 36.03 | 66.67 | 63.62 | 36.38 | 33.33 | 0.146 |
| **Mean value** | 70.43 | 29.26 | 76.91 | 69.94 | 30.06 | 23.09 | 0.259 |
| **S13** | 91.33 | 8.67 | 77.45 | 92.59 | 7.41 | 22.55 | 0.552 |
| **S14** | 89.12 | 10.88 | 75.49 | 90.36 | 9.64 | 24.51 | 0.480 |
| **S15** | 83.39 | 16.61 | 75.49 | 84.11 | 15.89 | 24.51 | 0.354 |
| **Mean value** | 87.95 | 12.05 | 76.14 | 89.02 | 10.98 | 23.86 | 0.462 |
| **S16** | 96.9 | 3.1 | 74.51 | 98.04 | 1.96 | 25.49 | 0.675 |
| **S17** | 94.55 | 5.45 | 74.51 | 95.51 | 4.49 | 25.49 | 0.527 |
| **S18** | 92.59 | 7.41 | 74.51 | 93.45 | 6.55 | 25.49 | 0.443 |
| **Mean value** | 94.68 | 5.32 | 74.51 | 95.67 | 4.33 | 25.49 | 0.548 |

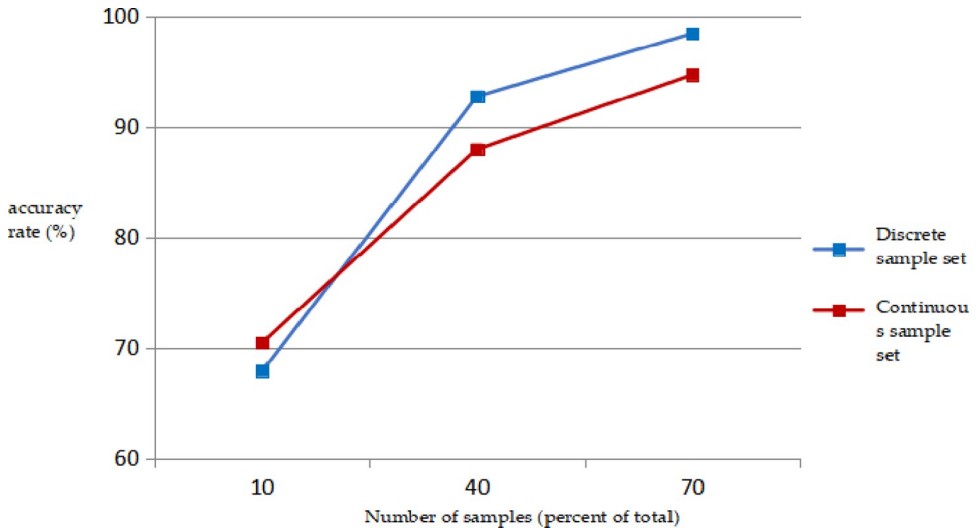

**Fig 1. Relationship between logistic regression prediction accuracy and number of samples.**

accuracy was more obvious. Taking S1, S2, and S3 as examples, although the number of three sample sets is 10% of the total number of samples, the prediction accuracies are not the same, that is, 75.38%, 64.12%, and 64.05%, respectively. The average precision and standard deviation are 67.85% and 6.52%, respectively. The number of sample sets of S7, S8, and S9 was also the same; both were 70% of the total, with a standard deviation of 0.42%.

The sensitivities of S14 and S15 are 75. 49%, respectively, and the sensitivities of S16, S17, and S18 is 74. 51%, indicating that the ability of the logistic regression model to correctly identify mineralization tended to be stable with an increase in the number of samples.

In addition, the number and type of samples also have an impact on the training results, as shown in Fig 1.

As shown in Fig 3 that when the number of sample sets was 10%, 40%, and 70% of the total, the average prediction accuracies of the discrete sample sets were 67.85%, 92.47%, and 98.42%, respectively, and the prediction accuracy of continuous sample sets is 70.43%, 87.95%, and 94.68%, respectively. The prediction accuracy of logistic regression also gradually increases, when the number of samples is 10% and 70% of the total, and the difference in prediction accuracy is close to 30% for both discrete and continuous sample sets.

When the number of sample sets is 10% of the total, the prediction accuracy of the continuous sample set is higher than that of the discrete sample set, but with an increase in the number of sample sets, the classification effect of discrete sample set is better than that of the continuous sample set; in general, the classification effect of the discrete sample set is slightly better than that of the continuous sample set [15–20].

In the process of mineral prediction, in addition to the overall accuracy, it is more important to correctly identify the positive samples, that is, to accurately identify metallogenic information. Therefore, the sensitivity of the model is an important indicator for measuring its classification effect. The variation in the sensitivity of the model is shown in Fig 2.

As shown in Fig 2 that when the number of sample sets was 10%, 40%, and 70%, of the total, the sensitivities of the discrete sample sets were 78.65%, 78.76%, and 70.59%, respectively, and the sensitivities of the continuous sample sets were 76.91%, 76.14%, and 74.51%, respectively. Instead of increasing, the sensitivity of the logistic regression decreased to varying degrees. When the number of samples was 70% of the total, the sensitivity decreased by 8%

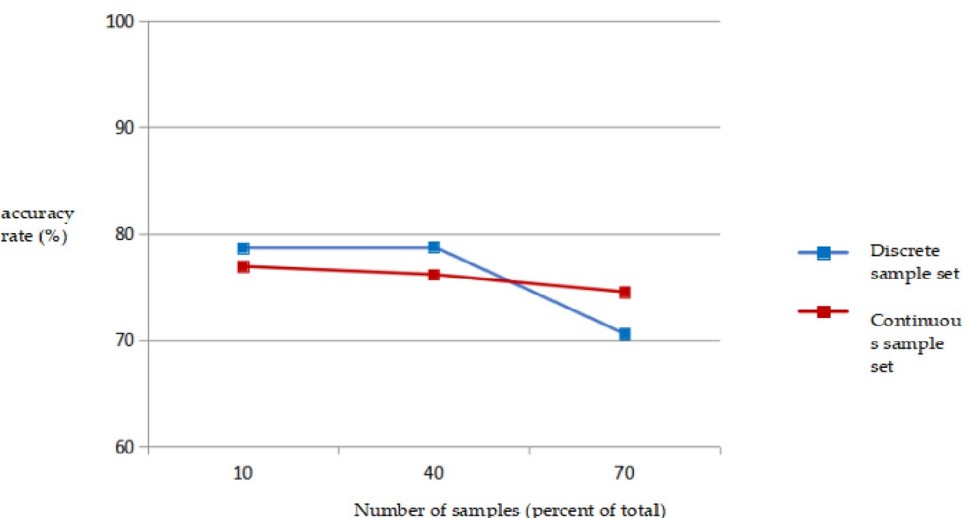

**Fig 2. Relationship between logistic regression prediction sensitivity and sample number.**

compared with 40%, indicating that the ability of the logistic regression model to correctly identify mineralization does not increase with an increase in the number of samples, but decreases to a certain extent.

When the number of training sample sets is the same, the sensitivity of discrete sample sets is close to that of continuous sample sets.When selecting the appropriate sample set for metallogenic prediction, the prediction accuracy and sensitivity need to be considered comprehensively. In the discrete sample set, the accuracy of S8 was the highest, reaching 98.75%; however, the sensitivity was only 72.55%. The sensitivity of S2 was the highest, (85.62%) but its accuracy was only 64.12%. The accuracy of S4 sample was 90.34%, and the sensitivity was 82.35%, which is the most suitable training sample in the discrete sample set. S13 was selected as the training sample in the continuous sample set [21].

The prediction degree curve is also an effective means of evaluating mineral prediction, which can intuitively reflect the efficiency of mineral prediction, that is, whether the minimum area can be used to predict the maximum number of mineral spots. The abscissa of the prediction degree curve is the cumulative area percentage of the study area and the ordinate is the percentage of the number of known ore spots predicted to the total number of ore spots.The prediction degree curve of the ideal prediction model should be an inverted "L" shape with an inflection point of (0,1), that is to say, the model delineates all known ore spots in a very small area, while the prediction degree curve of the poor prediction model is a diagonal line, that is to say, the number of known ore spots successfully predicted increases only with the increase of the area of the study area.It shows that the mineral prediction is the result of random classification. Therefore, the more the prediction degree curve deviates to the upper left, the higher the prediction efficiency of the model [22–26].

Using S4 and S13 as the training sample set, the mineralization probability of each unit grid in the study area was calculated using logistic regression, and the mineralization prediction degree curve was drawn, as shown in Fig 3.

As shown in Fig 3 that the prediction results for S4 and S13 were relatively close. However, the number of known mineral occurrences predicted in S4 is slightly greater than that in S13, so the metallogenic prediction map of the study area was drawn according to the prediction results of the S4 sample set, and the study area was divided into high potential area, medium

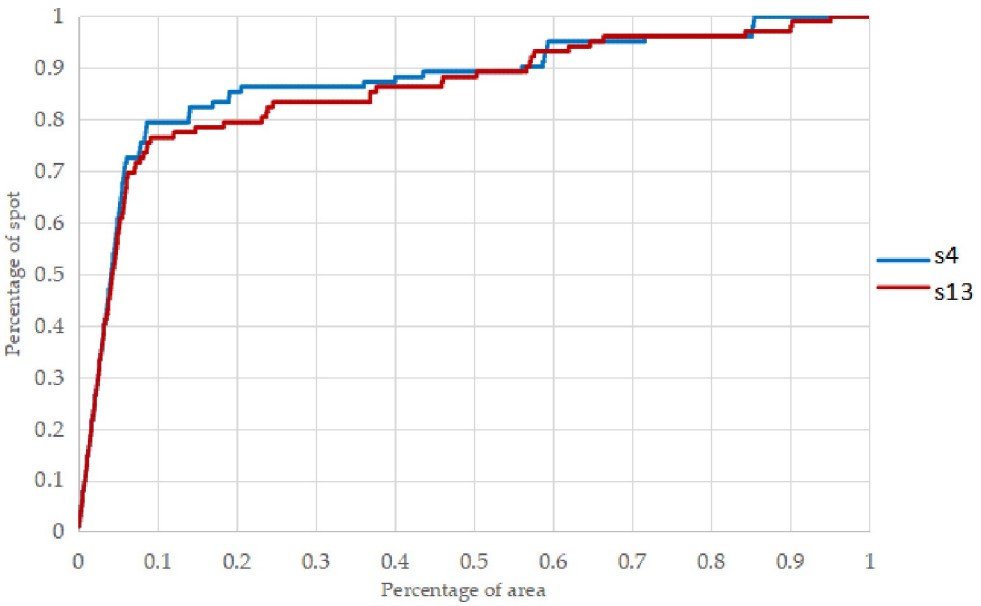

**Fig 3. Logistic regression predictability curve of sample set S4 and S13.**

potential area, and low potential area with 10% and 50% of the total area of the study area as the boundary.

the high mineralization potential area is mainly located in the Eastern, especially the clay layers, and is highly consistent with the known ore occurrences in this area, but the prediction effect of the known ore occurrences in the Southern Cross area is not ideal.According to statistics, the number of known ore occurrences predicted in the high potential area is 136, and the number of known ore occurrences predicted in the medium potential area is 17.

## 4. Mineral prediction model evaluation

In this experiment, the total number of known bauxite deposits in the training sample is 171. If 10% of the study area is considered to have a high potential area of mineralization, the numbers of known deposits predicted by the weight of evidence method is 94.However, the number of known mining points predicted by logistic regression, support vector, and random forest were 136, 144, and 164, respectively. It can be seen that the prediction ability of the machine learning algorithm is better than that of the evidence weight method.

Therefore, this section mainly uses classification accuracy, receiver operating characteristic curve, prediction degree curve and other indicators to comprehensively evaluate and optimize the three machine-learning algorithms of logistic regression, support vector machine and random forest.

### 4.1 Sample selection

Combined with the prediction results of logistic regression, support vector, and random forest, the influence of sample randomness, number, and type on the prediction results is discussed.

First, the prediction accuracy intervals and standard deviations of the three algorithms were compared when the number of samples was the same. The results are shown in Table 4.

As shown in Table 4 that the average prediction accuracy intervals of logistic regression, support vector, and random forest are 7.92, 12.36, and 7.15, respectively, and the standard

**Table 4. The prediction accuracy interval and standard deviation of the three algorithms.**

| | Average prediction accuracy interval | | | Standard deviation of average prediction accuracy | | |
|---|---|---|---|---|---|---|
| | LR | SVM | RF | LR | SVM | RF |
| S1–S3 | 11.33 | 15.49 | 18.58 | 6.52 | 8.57 | 9.29 |
| S4–S6 | 5.73 | 6.38 | 7.44 | 2.97 | 3.61 | 3.82 |
| S7–S9 | 0.8 | 0.9 | 6.16 | 0.41 | 0.45 | 3.50 |
| S10–S12 | 17.45 | 30.43 | 9.14 | 9.35 | 17.35 | 5.05 |
| S13–S15 | 7.94 | 20.37 | 1.23 | 4.09 | 10.38 | 0.66 |
| S16–S18 | 4.31 | 0.63 | 0.36 | 2.15 | 0.34 | 0.18 |
| Mean value | 7.92 | 12.36 | 7.15 | 4.24 | 6.78 | 3.75 |

deviations of the average prediction accuracy are 4.24, 6.78, and 3.75, respectively. It can be seen that when the number of samples is the same, the difference in random forest classification results is the smallest, followed by logistic regression, and the difference of support vector machine results is the largest Second, the impact of the number of samples on the three algorithms is compared and analyzed, as shown in Table 5.

We Figs 4 and 5 can more intuitively reflect the impact of the number of samples on different machine learning algorithms.

As shown in Fig 4 that with an increase in the number of samples, the prediction accuracy of the three machine learning algorithms increased to varying degrees, among which logistic regression had the largest increase.

As shown in Fig 5 that with an increase in the number of samples, the sensitivity changes of the three different machine-learning algorithms are different. The logistic regression shows a certain degree of decline, the support vector machine first rises and then falls, and the random forest increases slowly.

In general, only the prediction accuracy and sensitivity of random forest increase with an increase in the number of samples.

Finally, the effects of the4 sample types on the three algorithms are compared, as shown in Table 6.

Figs 6 and 7 can more intuitively reflect the impact of sample types on different machine learning algorithms.

It can be seen from Fig 6 that the accuracy of the logistic regression and support vector machine using the discrete sample set for metallogenic prediction is better than that of the continuous sample set, especially for support vector machine, and the difference in prediction

**Table 5. Impact of sample size on different machine learning algorithms.**

| Number of samples (%) | | Average prediction accuracy | | | Average sensitivity | | |
|---|---|---|---|---|---|---|---|
| | | LR | SVM | RF | LR | SVM | RF |
| 10 | S1–S3 | 67.85 | 86.15 | 80.24 | 78.65 | 64.71 | 81.48 |
| | S10–S12 | 70.43 | 78.05 | 77.72 | 76.91 | 50.33 | 84.1 |
| | Mean value | 69.14 | 82.1 | 78.98 | 77.78 | 57.52 | 82.79 |
| 40 | S4–S6 | 92.47 | 93.13 | 89.23 | 78.76 | 79.41 | 82.68 |
| | S13–S15 | 87.95 | 75.42 | 93.26 | 76.14 | 85.29 | 86.27 |
| | Mean value | 90.21 | 84.27 | 91.24 | 77.45 | 82.35 | 84.47 |
| 70 | S7–S9 | 98.42 | 97.56 | 93.45 | 70.59 | 73.86 | 81.04 |
| | S16–S18 | 94.68 | 97.71 | 95.83 | 74.51 | 68.63 | 92.16 |
| | Mean value | 96.55 | 97.635 | 94.64 | 72.55 | 71.245 | 86.6 |

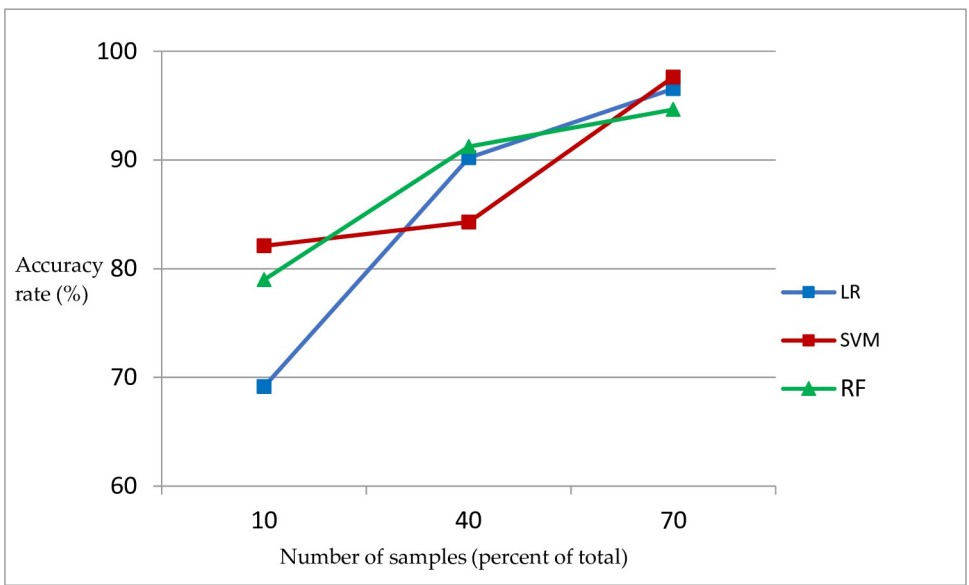

**Fig 4. Relationship between prediction accuracy of different machine learning models and the number of samples.**

accuracy between the discrete8 and continuous sample sets is close to 10%. However, the accuracy of the continuous sample set was better than that of the discrete sample set in the case of the random forest.

As shown in Fig 7 that there was little difference in the sensitivity of each machine learning algorithm using different types of training sample sets for metallogenic prediction. The sensitivity of the logistic regression was the same. The sensitivity of support vector machine using discrete sample sets is better than that of continuous sample sets, and the sensitivity of the random forest using continuous sample sets is better than that of discrete sample sets.

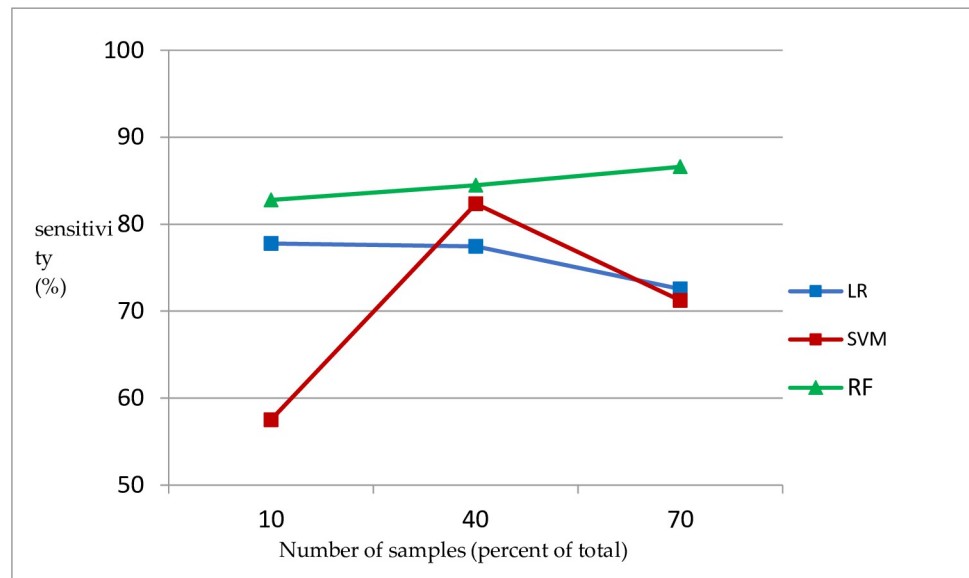

**Fig 5. Relationship between sensitivity of different machine learning models and number of samples.**

**Table 6. Impact of sample type on different machine learning algorithms.**

| Sample type | | Average prediction accuracy | | | Average sensitivity | | |
|---|---|---|---|---|---|---|---|
| | | LR | SVM | RF | LR | SVM | RF |
| Discrete | S1-S3 | 67.85 | 86.15 | 80.24 | 78.65 | 64.71 | 81.48 |
| | S4-S6 | 92.47 | 93.13 | 89.23 | 78.76 | 79.41 | 82.68 |
| | S7-S9 | 98.24 | 97.56 | 93.45 | 70.59 | 73.86 | 81.04 |
| | Mean value | 86.18 | 92.28 | 87.64 | 76 | 72.66 | 81.73 |
| Continuous | S10-S12 | 70.43 | 78.05 | 77.72 | 76.91 | 50.33 | 84.1 |
| | S13-S15 | 87.95 | 75.42 | 93.26 | 76.14 | 85.29 | 86.27 |
| | S16-S18 | 94.68 | 97.71 | 95.83 | 74.51 | 68.63 | 92.16 |
| | Mean value | 84.35 | 83.72 | 88.93 | 75.85 | 68.08 | 87.51 |

In general, logistic regression and support vector machines using discrete sample sets are better than continuous training sample sets for metallogenic prediction, whereas random forest is the opposite.

## 4.2 Prediction accuracy

The prediction accuracy evaluation of machine learning algorithms is mainly based on the confusion matrix, and the commonly used indicators include accuracy, misclassification rate, sensitivity, specificity, false positive rate, false negative rate, and the Kappa coefficient.

Table 7 shows the prediction accuracy of various machine learning models based on discrete sample set and continuous sample set respectively.

It can be seen from Table 7 that although the accuracy of logistic regression is more than 90%, its sensitivity is lower than that of other machine learning algorithms, whether based on discrete or continuous sample sets. Although the accuracy of random forest classification is not always the highest, the sensitivity of random forest is the best among the three machine learning algorithms, especially the random forest based on continuous samples, which has the best accuracy, sensitivity, and Kappa coefficient among all models.

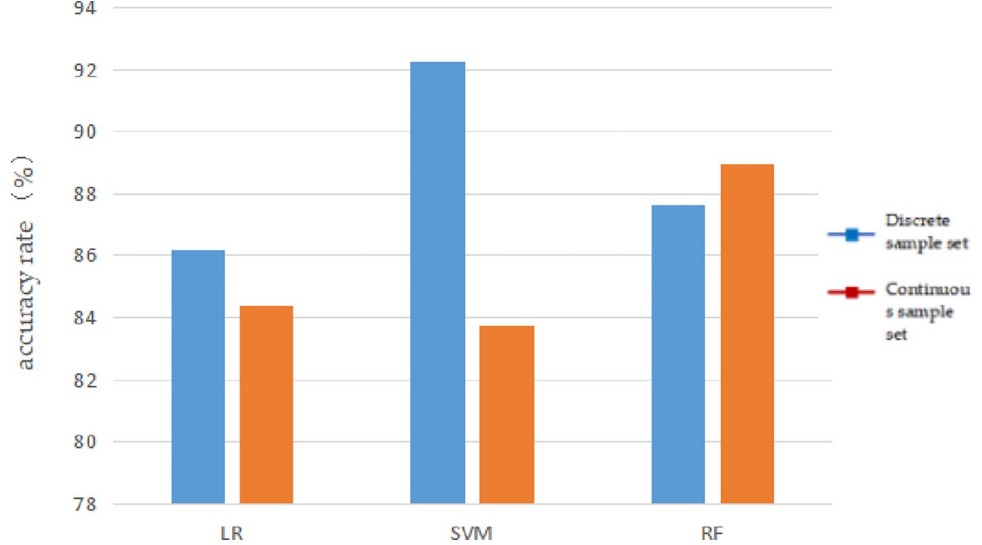

**Fig 6. Relationship between prediction accuracy of different machine learning models and sample types.**

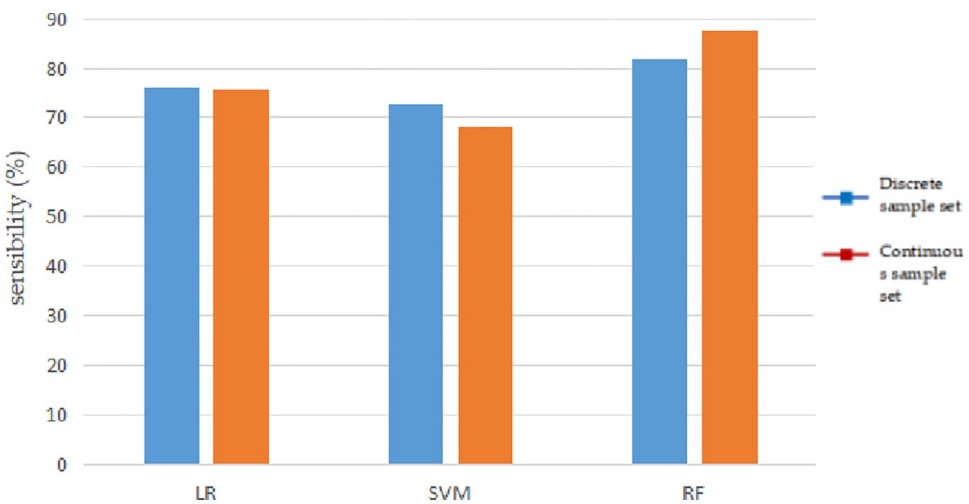

**Fig 7. Relationship between sensitivity of different machine learning models and number of samples.**

## 4.3 Receiver operating characteristic curve

When any classification model is used to solve a binary classification problem, it is necessary to set a threshold for the obtained classification probability. Taking 0.5 as example, if the classifier thinks that the probability of the sample belonging to the positive class is greater than 0.5, it is classified as the positive class; otherwise, it is classified as the negative class. If the threshold is reduced to 0.4, more positive classes can be correctly identified. The proportion of the positive class that is correctly identified to all positive classes is improved;that is, the sensitivity of the classifier is improved, but at the same time, more negative classes are classified as positive classes, that is, the false positive rate of the classifier is improved.

The receiver operating characteristic (ROC) curve is a visual classifier evaluation method, which is a coordinate graph composed of the false positive rate as the ordinate and the sensitivity as the abscissa to explain the relationship between the sensitivity and the false positive rate. In this coordinate system, the point (0, 0) indicates that both the sensitivity and the false positive rate are 0; that is, all samples are classified into negative classes, and the point (1,1) indicates that the sensitivity and the false positive rate are both 1; that is, all samples are classified into positive classes, the point (0, 1) indicates that the sensitivity is 1, and the false positive rate is 0, which indicates that all positive classes and negative classes are correctly classified, which is the most ideal classification state. Therefore the threshold corresponding to the point closest to the upper left corner is the best threshold for classification, and the classifier can reduce the negative classes into positive classes as much as possible.Identify as many positive classes as

**Table 7. Prediction accuracy of different machine learning models.**

| Model | Accuracy Misclassification rate | Sensitivity | Specificity | False | Positive rate | False negative rate | Kappa coefficient |
|---|---|---|---|---|---|---|---|
| Logistic regression with discrete samples | 90.34 | 9.66 | 82.35 | 91.07 | 8.93 | 17.65 | 0.227 |
| Logistic regression with continuous samples | 91.33 | 8.67 | 77.45 | 92.59 | 7.41 | 22.55 | 0.552 |
| Discrete sample support vector machine | 91.16 | 8.84 | 84.31 | 91.79 | 8.21 | 15.69 | 0.568 |
| Continuous sample support vector machine | 84.45 | 15.55 | 83.33 | 84.55 | 15.45 | 16.67 | 0.4 |
| Discrete sample random forest | 86.01 | 13.99 | 87.25 | 85.59 | 14.11 | 12.75 | 0.444 |
| Continuous sample random forest | 95.89 | 4.11 | 96.08 | 95.88 | 4.12 | 3.92 | 0.66 |

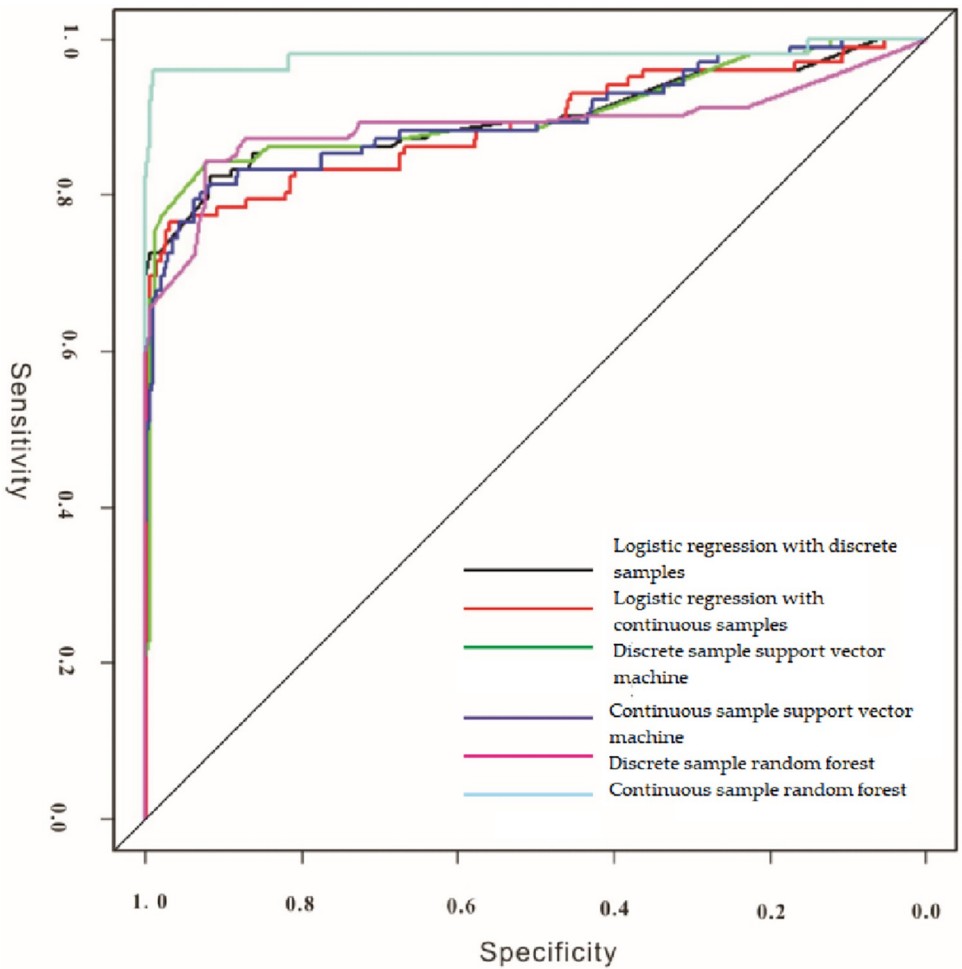

**Fig 8. ROC curve of different machine learning models.**

possible. When judging the classification effect of two or more classifiers, the ROC curves of different classifiers can be drawn in the same coordinate system, and the classifier corresponding to the ROC curve near the upper-left corner is the best.

Although the ROC curve is intuitive, it can not quantitatively describe the performance of the classifier. Therefore, the Area Under ROC Curve (AUC) was used to evaluate the classifier performance.

The AUC can use a single index of area instead of sensitivity and false positive rate to evaluate the classifier. When all the samples were classified correctly, the AUC of the classifier was 1. If a classifier is classified completely randomly, in theory, all the results of the classifier are on a diagonal line; that is, the AUC is 0.5. Therefore, if the AUC of a classifier is closer to 1, the classifier has a better classification effect. symmetric.

The receiver operating characteristic curves of different machine learning models were drawn according to,the confusion matrix of each machine learning model established previously, as shown in Fig 8.

Results of short-and long-run asymmetric results proposed by Diks and Panchenko. Table 8 shows the linear Granger causality test for economic growth, electricity consumption, and carbon emissions.

**Table 8. AUC area of different machine learning models.**

| Machine learning model | AUC area |
|---|---|
| Discrete sample logistic regression model | 0.901 |
| Continuous sample logistic regression model | 0.8904 |
| Discrete sample support vector machine model | 0.9034 |
| Continuous sample support vector machine model | 0.8982 |
| Discrete sample random forest model | 0.8901 |
| Continuous sample random forest model | 0.979 |

Using the plotted ROC curves, the areas under the ROC curves for the different machine learning models were calculated, as shown in Table 8.

As shown in Table 8 that the AUC area of the random forest based on continuous samples is the largest, reaching 0.979, indicating that the random forest can identify more information on mineral occurrences in the study area, while reducing the misclassification of non-mineral information into mineral occurrences. The second was logistic regression based on discrete samples, but the AUC area of logistic regression based on discrete samples was only 0.901.The results show that the classification effect is different from that of a random forest based on continuous samples. The AUC areas of the other machine learning algorithms were close to those of logistic regression based on discrete samples.

In addition, it can be seen that the AUC area of random forests based on continuous samples is significantly larger than that of random forests based on discrete samples. However, the classification results of the other two algorithms using discrete samples were better than those using continuous samples, which indicates that different machine learning algorithms are not suitable for the same type of samples [27–32].

## 4.4 Prediction degree curve

The prediction degree curve is an important index for evaluating metallogenic prediction, and mainly reflects the efficiency of the metallogenic prediction model, that is, whether it can predict as many ore spots as possible with as small an area as possible. The prediction degree curve takes the cumulative area percentage of the study area as the abscissa and the percentage of the predicted number of ore occurrences to the total number as the ordinate. As with the ROC curve,the model corresponding to the prediction degree curve closest to the upper-left corner has the best metallogenic prediction effect.

The prediction degree curves of the different models are drawn according to the prediction results of each machine learning algorithm, as shown in Fig 9.

It can also be seen from the prediction curve that the prediction effect of the random forest based on continuous samples is significantly better than that of other algorithms, the prediction effect of support vector machine based on continuous samples is the worst, and the prediction effects of other algorithms are relatively close.

In addition, the number of known ore occurrences contained in the high potential area in the prediction results of different algorithms was also counted, as shown in Table 9.

As shown in Table 9 that the high-potential mineralization area predicted by random forest based on continuous samples contained the largest number of mineral occurrences, up to 164, accounting for 95.9% of the total number of mineral occurrences, followed by the support vector machine based on discrete samples, with 144 known mineral occurrences predicted, accounting for 84.2% of the total number. The total number of known occurrences predicted by other algorithms was less than 80% of the total number of occurrences.

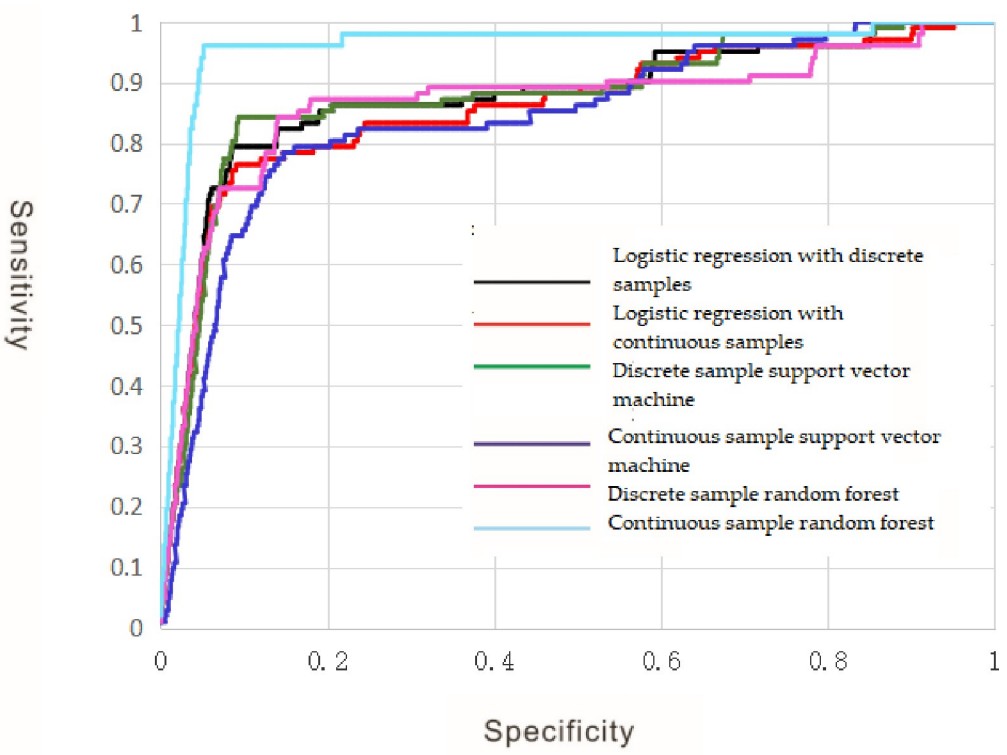

**Fig 9. Prediction degree curves of different machine learning models.**

## 4.5 Assessment of prospecting factors

n metallogenic prognosis, multiple prospecting factors are usually used, and the influence of each prospecting factor on metallogenesis is not the same, so how to evaluate the importance of each prospecting factor, find out the geological elements with the greatest correlation with metallogenesis, and eliminate the elements with no correlation with metallogenesis is of great significance for metallogenic prognosis.

In this study, the importance of prospecting factors was ranked using the Random Forest Recursive Feature Elimination (RF-RFE) method.

The basic idea of recursive feature elimination is to repeatedly construct the learning model, then select the least important features, put forward the selected features, and then repeat the previous process on the remaining features until all the features have been traversed. In this process, the order in which features are eliminated is the order of importance of the features.

**Table 9. Mineral occurrences predicted by different models in high potential mineralization areas.**

|  | Number | Premeasure |
|---|---|---|
| Machine learning model | 136 | 79.5 |
| Discrete sample logistic regression model | 131 | 76.6 |
| Continuous sample logistic regression model | 144 | 84.2 |
| Discrete sample support vector machine model | 112 | 65.5 |
| Continuous sample support vector machine model | 124 | 72.5 |
| Discrete sample random forest model | 164 | 95.9 |

When the RF-RFE feature selection method is applied, the misclassification rate of each feature variable is recalculated in each iteration process. Thus, the influence of the elimination factor on the combination of the remaining variables can be avoided, and the result is more reliable.

Based on RF-RFE feature selection method,The importance of prospecting factors in the study area is listed as follows:clay rock > aluminum anomaly > iron oxide content > the first principal component anomaly > fault structure > ratio of alumina to iron oxide > gallium anomaly > lithium anomaly > scandium anomaly. However, according to the analysis results of the evidence weight model, the importance order of the prospecting factors is as follows: clay rock > ratio of alumina to iron oxide content > aluminum anomaly > fault structure > gallium anomaly > lithium anomaly > scandium anomaly > first principal component anomaly > iron oxide content. Both methods show that claystone is the most important ore-controlling factor for bauxite deposits, but the importance of other prospecting factors is slightly different. For example, the evidence weight analysis shows that the five geochemical elements anomalies of Li, Sc, Fe, Ga and Cr have the weakest correlation with mineralization, while the RF-RFE method shows that the importance of Al and Li elements is second only to claystone. It can be seen that RF-RFE method can more accurately reflect the relationship between the prospecting factors and the spatial distribution of the deposit [33–35].

## 5. Application cases

Based on the predicted results, we conducted a geological survey in Wuchuan County and discovered the Yanfengqian bauxite deposit. The details are as follows:

### 5.1 Geography of mining area

The Yanfengqian Bauxite Mine is located in the north of Zhuoshui Town, Wuchuan County, Guizhou Province, and is under the jurisdiction of Zhuoshui Town. Geographic extreme coordinates: east longitude: 107°54′00″~107°58′30″; North latitude: 28°58′45″~ 29°01′45″. It is 90 km from the mining area to Wuchuan County and 320 km from Zunyi County. The highway from Wuchuan to Wulong County passes through the west side of the mining area, and the highway from Zhuoshui to Pengshui County via Luchi passes through the middle part of the mining area, so traffic is relatively convenient.

The mining area is located at the junction of Guizhou and Chongqing in the eastern branch of Dalou Mountain, the main mountain range in northern Guizhou, and is the watershed of the Furong and Hongdu rivers, the main stream of the Wujiang River system. The karst landform developed in the mining area is a plateau and synclinal platform landform. The general elevation above sea level is 1000–1300 meters, the highest elevation is 1444.6 meters (about 780 meters in the west of the mining area), the lowest elevation above sea level in the mining area is 618.2 meters (about 650 meters in the southwest of Xiongjiagou), the relative height difference is 826.4 meters, and the lowest erosion surface elevation is 618.2 meters. It is a shallow-to medium-cut terrain from the middle and low mountains to the middle mountains. The landform at the edge of the mining area fluctuates greatly, while the wing of the syncline is a plateau platform formed by Permian and Triassic limestone. The core of the syncline is often a gentle slope, hills, and sometimes a small intermountain basin. Carbonate rocks are widely distributed in mining areas, and karst landforms such as underground rivers, karst caves, funnels, and isolated peaks are relatively developed. The east and west of the mining area are overhanging cliffs or steep slopes formed by Middle Permian limestone or Upper Permian limestone above the aluminum-bearing rock series with a general height of 150–250 meters and a maximum height of 325 m.

The mining area has a mid-subtropical humid monsoon climate, with an annual average temperature of 15.6˚C, the highest temperature of 39.5˚C, and the lowest temperature of -6.8˚C. Rainfall is concentrated from April to September and accounts for 76% of the annual rainfall. Owing to the special terrain, it is often overcast and rainy from September to April of the following year, and the climate is cold.

The mining area is located in an area where the seismic intensity is less than VI, and the stability is good.

There is no large river or surface water body in the mining area, and there is only one seasonal gully from Yanjiao to Xijiaba, which mainly gathers surface water on the two wings of the syncline. The surface water flows from Yanjiao to the sinkhole near Xijiaba, which is dry year-round, and mainly drains the surface water on the two wings of the syncline during the flood period. Occasionally, some small spring wells can meet the needs of local villagers during normal years.

The main fuel in the mining area is coal, which originates from Nigao and Shichao in Wuchuan County and Fengle in the south.

Agriculture is currently the pillar industry in mining areas, and the main food crops are corn, wheat, and rice. Followed by potatoes and sweet potatoes, cash crops are flue-cured tobacco, rapeseed, tea, sericulture, raw lacquer, etc.

Domestic water in the mining area is mainly extracted from karst caves and springs, which can meet the living needs of local residents and workers; industrial water can be developed from the tributaries of the Furong and Hongdu River.

## 5.2 Resource estimation

Based on the occurrence characteristics and occurrence law of the ore body, combined with factors such as the strata of the mining area, the traditional "exploration line method" was selected for exploration. Drilling engineering is used to control the deep ore body, supplemented by trenching, shallow wells, and stripping to expose the shallow ore body and the aluminum-bearing rock series. Surface mapping was carried out on the 1:10000 topographic map, and the ore body outcrop and other stratigraphic and structural boundaries were controlled at fixed points by using the crossing method as the main method and the tracing method as the auxiliary method, In which, the basic analysis sample of bauxite was systematically taken to delineate the ore body. In the process of exploration, the detailed survey design and the principle of "construction, integration, and adjustment at the same time" are followed to make the work arrangement more reasonable and effective.

The results show that the deployment of geological work is reasonable, and the working methods and means used to control the distribution, occurrence, scale of the ore body and identify the quality of the ore have reached the corresponding working level.

It is estimated that the resources of 332 + 333 in Yanfengqian Bauxite Mine are 28.9503 million tons, including 3.3664 million tons of 332 resources, accounting for 11.63% of the total resources in the mining area, and 25.5839 million tons of 333 resources, accounting for 88.37% of the total resources in the mining area.(Table 10)

The industrial grade of gallium (Ga) is 0.0020%, that of lithium (Li) is 0.032%, and that of scandium (Sc) is 0.0020%. According to the statistics of the composite sample of Yanfengqian bauxite, the average content of the associated beneficial components in the bauxite ore was 0.0032% for gallium (Ga), 0.0057% for lithium (Li), and 0.0022% for scandium (Sc), and the average contents of gallium (Ga), lithium (Li) and scandium (Sc) are higher than the industrial grade.

The estimation of Ga, Li, and Sc resources is based on the product of their average content and bauxite ore quantity. Because the average contents of gallium, lithium, and scandium were

**Table 10. Estimation results of bauxite resources in Yanfengqian.**

| Block | True Area (m2) | Average thickness (m) | Volume (m3) | Weight (t/m3) | Resource reserves (10,000 tons) | | | Ratio (%) |
|---|---|---|---|---|---|---|---|---|
| Number | | | | | 332 | 333 | Totally | Proportion |
| ① | 573698.08 | 1.98 | 1135922.21 | 2.67 | 336.64 | | 2895.03 | 11.63 |
| ② | 1959487.38 | 2.02 | 3958164.51 | 2.67 | | 1056.83 | | 36.53 |
| ③ | 68239.82 | 1.03 | 70287.01 | 2.67 | | 18.77 | | 0.65 |
| ④ | 3355086.09 | 1.69 | 5670095.49 | 2.67 | | 1513.92 | | 52.29 |
| | Totally | | | | 336.64 | 2558.39 | | 100 |

calculated from the analysis results of the combined samples, they were not divided into separate blocks; therefore, the resource category was classified as 333.

The mining area is estimated to contain 926.41 tons of gallium (Ga), 6051.23 tons of lithium oxide and 636.91 tons of scandium (Sc). See Table 11 for the estimation results.

## 5.3 Economic benefit calculation

The Yanfengqian bauxite deposit in Wuchuan County was planned as the reserve resource guarantee area for the aluminum industrial base in northern Guizhou. The economic significance of deposit development is evaluated by the total profit method, and the technical and economic indexes selected for the estimation of its economic utilization value are as follows: recoverable bauxite resources (Qs), conversion ratio of ore required to produce 1 ton of alumina (A), and the expected total profit (I) during the entire mining period of the deposit. The formulas for calculating the above three technical and economic indicators and the adopted parameters are listed below.

1. Minable bauxite resources (Qs):

    Qs = Q·K·ε·(1 + ρ) = 2895.03*0.7*0.8*1.06 = 171.849 (ten thousand tons)
    Where:
    Q-332 + 333 ore resources (28.9503 million tons)
    K-coefficient of recoverable resources within the mining boundary (0.7)
    ε-bauxite mining recovery rate (80%)
    ρ-Ore dilution rate (6%)

2. Conversion ratio of ore required to produce 1 ton of alumina (A):

$$A = \frac{\beta A}{\beta \cdot \varepsilon A(1 - \rho)} = \frac{0.986}{0.6124 \times 0.8 \times (1 - 0.06)} = \frac{0.986}{0.4605} = 2.14$$

   Where:
   βA-Average content of $Al_2O_3$ in alumina product (98.6% for first-class product in Bayer process)
   Average content of $Al_2O_3$ in β-bauxite ore (61.24%)

**Table 11. Estimation results of associated mineral resources of Yanfengqian bauxite.**

| Associated minerals | Ore quantity of bauxite in the mining area (10,000 tons) | Average grade of gallium in the mining area (%) | Associated Mineral Resources (t) |
|---|---|---|---|
| Gallium | 2895.03 | 0.0032 | 926.41 |
| Lithium | 2895.03 | 0.0057 | 6051.23 |
| Scandium | 2895.03 | 0.0022 | 636.91 |

εA-comprehensive recovery rate of alumina (80%)

ρ-Ore dilution rate (6%)

According to the current situation and prediction of the domestic aluminum industry development, the geological characteristics of the deposit, the current situation of resources, and its burial conditions, the expected production scale of the alumina plant can be determined as 400,000 tons of alumina per year (F1), and the ore demand is approximately 860,000 tons per year (F2).

Mine service life (T): T = Qs/F2 = 17,184,900 tons /860,000 tons/year = 20 years

3. Expected total profit (I) during the whole mining period of the deposit

$$I = Qs\{\frac{P-C}{A} - \frac{C2}{A \cdot T} - \frac{C1}{T}\} - E = 1718.49\{\frac{2900-1500}{2.14} - \frac{3500}{2.14 \times 20} - \frac{500}{20}\} - 2430$$

= 938323 (ten thousand yuan)

Where:

P-Price of alumina: According to the quotation of Chinalco, the quotation of Chinalco is 2900 yuan/ton (after-tax price).

C-Production cost of alumina (including mining, transportation, crushing, financial and management expenses, resource tax, etc., and the cost of alumina production by Bayer process), which is 1500 yuan/ton according to relevant data.

C1-Capital construction investment of bauxite of mine unit (500 yuan/ton).

C2-Investment in alumina plant (3500 yuan/ton).

T-mine service life (20 years).

E-Geological exploration investment (total investment of 24.3 million yuan).

## 6. Discussion

The evidence weighting method itself has certain limitations, including the requirement of the input data to follow a certain distribution pattern and the need for conditional independence between input features. To overcome these problems, many scholars have proposed improved methods for the evidence weight method, such as Agterberg's weighted evidence weight model based on the logistic regression method, which uses logistic regression coefficients as weighting factors to obtain a mixed calculation model of logistic regression and evidence weight method, effectively solving the problem of maintaining the conditional independence of each evidence layer in the evidence weight model; Cheng Qiuming proposed a theoretical model for enhancing evidence rights, and compared it with ordinary evidence rights methods through examples. The results showed that the enhanced evidence rights method significantly improved the practical application effect; Sun Tao et al. used the evidence weight method combined with fractal method to delineate the tungsten mineralization prospects in the southern Jiangxi region, and achieved good prediction results. Commonly used shallow machine learning methods in the field of mineralization prediction include support vector machines, random forests, and artificial neural networks. Zuo et al. applied a support vector machine algorithm to predict mineralization prospects. Research has shown that the support vector machine algorithm integrates multiple evidence variables for mineralization prospect prediction, and its results are significantly better than those of the evidence weight method. Du et al; Wang et al. used the semi supervised random forest method to map the mineralization prospects of the southwestern Fujian metallogenic belt, indicating that semi supervised learning can improve the effectiveness of mineralization prospect mapping; Li et al. used the random forest algorithm based on bagging technology to predict the metallogenic prospect of tungsten

polymetallic deposits in the the Nanling Mountain metallogenic belt, which overcame the uncertainty caused by the discrete evidence map and the lack of data caused by the scarcity of deposits/ore occurrences. In recent years, with the maturity of machine learning in the field of mineralization prediction, coupled with the rapid growth of geological data and the continuous improvement of computer technology, a large number of experts and scholars have pushed deep learning-based mineralization prediction into a research boom. Ghezelbash used two supervised machine learning algorithms-a radial basis function neural network and a support vector machine with an RBF kernel, to generate a data-driven porphyry copper deposit prediction model. The results showed that the radial basis function neural network had the best performance in delineating the ore-forming prospect areas. Machine learning algorithms have a strong ability to handle nonlinear spatial relationships and can integrate massive amounts of multi-source spatial data. There are various types of machine learning algorithms, and different algorithms exhibit inconsistent predictive performances in different regions. Therefore, considering the multi-source, heterogeneous, and nonlinear characteristics of geological big data in the current era, this study combines previous research and fully explores multi-source geological information. Based on this, shallow machine learning algorithms, such as random forest and neural networks, as well as typical deep learning algorithms, are combined with geological big data to carry out tungsten mineralization prediction work in the study area, aiming to seek the best prediction algorithm in the study area. In addition, we obtained certain results.

## 7. Conclusions

Weights of evidence and three different machine-learning algorithms were used for mineral prospecting in this study. The main outcomes include the following 4 aspects:

1. Discrete and continuous samples were used as training samples to evaluate the prediction accuracy of each machine learning algorithm; continuous samples (the grid value of each predictor map is a series of 0 and 1, and the discretization criterion is whether there is any anomaly) are suitable for logistic regression and support vector machine; and discrete samples (the grid value of each predictor map is the original sample value) perform better for random forest.

2. The confusion matrix, receiver operating characteristic (ROC) curve, and predictive-efficiency curves were used to evaluate the prediction results of each machine algorithm. Random Forest based on continuous training samples outperformed other algorithms in metallogenic prediction, and 95.9% of the deposits were located in the high prospective zones which occupied 10% of the total area. Weights of evidence and random forest recursive feature elimination were applied to assess the importance of each prospecting factors. The outcomes were not completely consistent with each other. Both results show that clay rock and aluminum anomalies are the most requisite. Thus, the RF-RFE method can more effectively reveal the spatial relationships between the prospecting factors and ore deposits.

3. It can also be seen that most of the bauxite deposits in the region, including all well-known deposits such as Dazuyuan, Xinhua, and Luolobng, are located in the high potential area of mineralization. This shows that the predicted results are in good agreement with known ore occurrences. In addition, there are still some high -prospectivity areas where no known deposits have been found and can be treated as key areas for future exploration.

4. According to the predicted results, after on-site exploration, design, and construction, Yanfengqian bauxite deposit was discovered, with an average thickness of 1.82 meters; The average content of $Al_2O_3$ is 61.24%; The resource amount is 28.9503 million tons.the

current situation and prediction of the domestic aluminum industry development, the geological characteristics of the deposit, the current situation of resources and its burial conditions, the expected production scale of the alumina plant can be determined as 400,000 tons of alumina per year, and the ore demand is about 860,000 tons per year. The service life of the mine is 20 years, with great economic potential and development value.

5. Previous ore-forming prediction work in the northern Guizhou region mainly focused on qualitative research. Applying quantitative methods to the exploration of bauxite deposits in the northern Guizhou region can provide a portable system and quantitative model support for future ore-forming predictions in the region. At the same time, clay layer anomalies have been overlooked in previous research and exploration work. The weights of evidence and machine learning prediction models in this study consistently indicate that clay anomalies contribute only to mineralization prediction, second only to aluminum anomalies, and are one of the most explicit prospecting indicators at the regional scale. This discovery can provide important new ideas and clues for prospecting and exploration in this area as well as new evidence for the contribution of regional surrounding rocks to bauxite mineralization.

## 8. Shortcomings and future work prospects

Weights of evidence and three different machine-learning algorithms were used for mineral prospecting in this study. The main outcomes include the following 4 aspects.

Information related to mineralization must be comprehensive. Deep prediction is the focus of future mineral exploration work, and the quantitative prediction research approach in this study is also applicable to deeper resource exploration. However, additional layers of deep mineralization information need to be added to transform the planar prediction problem in this study into three-dimensional prediction. From the analysis of the prediction results in this article, it can be seen that the contribution of geophysical information reflecting deep elements to the prediction results is the worst, indicating that there are certain deficiencies in the resolution and indicative nature of the deep mineralization information layer collected in this study at the regional (local) mining area scale.

The insufficient sample size in the training set is a challenge for quantitative mineralization prediction. Mineral deposits/occurrences are scarce, and obtaining reliable prediction results using limited samples is an important direction for future work. Robust sample expansion algorithms and reliable expansion verification mechanisms can assist in solving this problem.

The important contribution of clay and carbonate rock anomalies to mineralization prediction is a major discovery of this study. The theoretical analysis the spatial and genetic relationship between the surrounding rock and bauxite formation is of great research significance, but it is beyond the scope of this article. In the future, the use of spatial quantitative analysis combined with deposit geochemistry, micro-area observation techniques, and numerical simulation of mineralization may achieve breakthroughs in this regard. In addition, further exploration of other geochemical information related to bauxite mineralization will be carried out in the next step of this work.

## Author Contributions

**Conceptualization:** Tao Cui.

**Resources:** Min Zhang.

**Software:** Min Zhang.

**Supervision:** Min Zhang.

**Validation:** Shiqiang Yang.

**Visualization:** Shiqiang Yang.

**Writing – original draft:** Shiqiang Yang, Wu Yang.

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
