## [Decision Letter · Decision Letter 0]

10 Mar 2024

PONE-D-24-02196Prediction and practical application of bauxite mineralization in Wuzhengdao area, Guizhou, ChinaPLOS ONE

Dear Dr. Yang,

Thank you for submitting your manuscript to PLOS ONE. After careful consideration, we feel that it has merit but does not fully meet PLOS ONE’s publication criteria as it currently stands. Therefore, we invite you to submit a revised version of the manuscript that addresses the points raised during the review process.

**Dear Authors,**

The evaluations from the peer reviewers regarding your submitted work have been duly received. Upon reviewing their feedback, it is evident that they recommend that you **revise** your manuscript. Therefore, the authors should consider each comment and decide on the best course of action for their research.==============================

We look forward to receiving your revised manuscript.

Kind regards,

Shaker Qaidi

Academic Editor

PLOS ONE

“This research was supported by the Science and Technology Support Plan of Guizhou Province[Qian(2017)1410]), the High-level talent introduction program for the Guizhou Institute Of Technology (0203001018040),Special Projects and Research Topics of Guizhou Association for Science and Technology(0201011022001)”

5. We note that Fig.3-4 in your submission contain [map/satellite] images which may be copyrighted. All PLOS content is published under the Creative Commons Attribution License (CC BY 4.0), which means that the manuscript, images, and Supporting Information files will be freely available online, and any third party is permitted to access, download, copy, distribute, and use these materials in any way, even commercially, with proper attribution. For these reasons, we cannot publish previously copyrighted maps or satellite images created using proprietary data, such as Google software (Google Maps, Street View, and Earth). For more information, see our copyright guidelines: http://journals.plos.org/plosone/s/licenses-and-copyright.

a. You may seek permission from the original copyright holder of Fig.3-4 to publish the content specifically under the CC BY 4.0 license. 

Reviewers' comments:

Reviewer's Responses to Questions

**Comments to the Author**

1. Is the manuscript technically sound, and do the data support the conclusions?

Reviewer #1: Yes

Reviewer #2: Yes

2. Has the statistical analysis been performed appropriately and rigorously? 

Reviewer #1: Yes

Reviewer #2: I Don't Know

3. Have the authors made all data underlying the findings in their manuscript fully available?

Reviewer #1: Yes

Reviewer #2: Yes

4. Is the manuscript presented in an intelligible fashion and written in standard English?

Reviewer #1: Yes

Reviewer #2: Yes

5. Review Comments to the Author

Reviewer #1: The researchers conducted a study using machine learning algorithms to predict ore economic data under low data quantity or low reliability data conditions.

The rationale for the methods used were discussed following the literature review. In this context, it can be said that the objective-method connection of the study has been established correctly.

The theoretical basis of the applied methods and the field of study, as well as the data structure of the study, are well defined. The produced data were discussed in detail and the findings of the analyses were linked to the data.

The algorithms used in the study and the implementation steps are explained in detail. The result of the study showed that the RF-RFE algorithm produced consistent results.

Reviewer #2: (1) The paper is well-organized, but you may consider breaking down some long paragraphs into smaller ones for better readability.

(2) Ensure consistency in terminology and use of acronyms throughout the paper.

(3) Consider providing a more concise and focused introduction that clearly outlines the research problem, objectives, and significance.

(4) Specify the research gap or novelty that your study addresses.

(5) Expand the literature review to include recent and relevant studies in the field, emphasizing the gaps your research aims to fill. For instance

a. Mitigating tribological challenges in machining additively manufactured stainless steel with cryogenic-MQL hybrid technology

b. Parallel structure of crayfish optimization with arithmetic optimization for classifying the friction behaviour of Ti-6Al-4V alloy for complex machinery applications

c. A synergy of an evolutionary algorithm with slime mould algorithm through series and parallel construction for improving global optimization and conventional design problem

d. Pelton wheel bucket fault diagnosis using improved shannon entropy and expectation maximization principal component analysis

e. An adaptive feature mode decomposition based on a novel health indicator for bearing fault diagnosis

(6) Provide more context on the specific challenges or issues addressed by your research.

(7) Provide more detailed insights into the findings, explaining the practical implications of different motion states observed.

(8) Consider including more visual aids (charts, graphs) to enhance the presentation of results.

(9) Summarize the key findings in a concise manner, emphasizing their significance and practical implications..

(10) Consider explicitly stating how your findings contribute to the existing knowledge in the field.

(11) Include a section on potential future work or research directions based on the current findings.

6. PLOS authors have the option to publish the peer review history of their article (what does this mean?). If published, this will include your full peer review and any attached files.

Reviewer #1: No

Reviewer #2: No

---

## [Author Response · Author response to Decision Letter 0]

26 Mar 2024

Dear editor：

Thank you very much for your letter. We have learned much from your and two reviewers’ comments, which are fair, encouraging and constructive. After carefully studying the comments and your advice, we have made corresponding changes. The main revisions are listed below.

For reviewer one:

1.The researchers conducted a study using machine learning algorithms to predict ore economic data under low data quantity or low reliability data conditions.

The rationale for the methods used were discussed following the literature review. In this context, it can be said that the objective-method connection of the study has been established correctly.

 The theoretical basis of the applied methods and the field of study, as well as the data structure of the study, are well defined. The produced data were discussed in detail and the findings of the analyses were linked to the data.

The algorithms used in the study and the implementation steps are explained in detail. The result of the study showed that the RF-RFE algorithm produced consistent results.

. Thanks to the opinions of the reviewer, .There is an objective fact that the data related to minerals are relatively small, and there must be randomness, which requires us to interpolate numbers. In geological exploration, due to the deep burial of mineral resources and complex geological conditions, mineral exploration work is highly risky and the probability of success is not high. Often. Even for mines that have already been mined, it is impossible to accurately determine the true situation of the ore body. Finding complex and ever-changing ore bodies requires collaborative work from all parties to achieve. Simulation and robot learning are a good choice, and indeed, we fully agree with the "low data quantity or low reliability data conditions" mentioned by the judges in the article. Mineral data does have this characteristic, and our work only increases the probability of mineral exploration, even if it increases from 1% to 2%. From the results, it seems fortunate that we found the mine in the drilling construction verification stage, which is a combination of theory and practice.At the same time, we have also summarized the shortcomings of this work, and in the next step of work, we will continue to recommend this work

Shortcomings and Future Work Prospects 

(1)The information related to mineralization needs to be comprehensive. Deep prediction is the focus of future mineral exploration work, and the quantitative prediction method research approach in this article is also applicable to deeper resource exploration. However, more layers of deep mineralization information need to be added to transform the planar prediction problem in this article into three-dimensional prediction; From the analysis of the prediction results in this article, it can be seen that the contribution of geophysical information reflecting deep elements to the prediction results is the worst, indicating that there are certain deficiencies in the resolution and indicative nature of the deep mineralization information layer collected in this article at the regional (local) mining area scale. 

(2)The insufficient sample size in the training set is a challenge in quantitative mineralization prediction. Mineral deposits/occurrences are scarce, and obtaining reliable prediction results through limited samples is an important direction for future work. Robust sample expansion algorithms and reliable expansion verification mechanisms may provide assistance in solving this problem.

 (3)The important contribution weight of clay anomalies and carbonate rock anomalies to mineralization prediction is a major discovery of this article. How to theoretically reveal the spatial and genetic relationship between surrounding rock and bauxite formation is of great research significance, but it goes beyond the scope of this article. In the future, the use of spatial quantitative analysis combined with deposit geochemistry, micro area observation techniques, and numerical simulation of mineralization may achieve breakthroughs in this issue. In addition, further exploration of other geochemical information related to bauxite mineralization can be carried out in the next step of work.

For reviewer two:

1 The paper is well-organized, but you may consider breaking down some long paragraphs into smaller ones for better readability.

We have made some adjustments and deletions to the structure of the article

2 Ensure consistency in terminology and use of acronyms throughout the paper.

Thank you for your suggestion. This is very important. We have checked the terminology and abbreviations in the paper and made some modifications. If there are any that have not been fully revised, please help us point them out and we will make corrections again

3 Consider providing a more concise and focused introduction that clearly outlines the research problem, objectives, and significance.

We have adjusted the content of the introduction section to highlight research problems, objectives, and significance：Mineral resources are an important component of natural resources and a crucial material foundation for the development of human society. Mineral resource prediction and evaluation are scientific predictions and comprehensive evaluations of the possible locations and resource potentials of economically valuable minerals on Earth. In the 1950s, a team led by French scholar Allais conducted the first systematic and decision supportive mineral resource evaluation work in the Sahara Desert region of Algeria, North Africa, opening a new chapter in quantitative prediction of mineral resources. Machine learning is currently widely used in mineral resource prediction and evaluation systems, playing an important role in extracting and integrating predictive element information. Bayesian theorem, least squares method, and Markov chain method are widely used techniques in machine learning today. Therefore, from this perspective, machine learning actually developed several centuries ago, and classic methods in mineral prediction, such as evidence weight method and logical regression, should also belong to the category of machine learning. Since the 20th century, especially since Alan Turing proposed the establishment of the first learning machine in 1950, deep learning has been widely applied in practical applications such as facial recognition, speech recognition, speech translation, etc. in the early 21st century, and modern machine learning has made significant progress. Overall, machine learning is learning a certain pattern or model from data and using it to solve practical problems. It is good at handling nonlinear and high-dimensional data, and is widely used in many disciplinary fields. Since the 1970s, when mineral prediction entered the quantitative stage, machine learning and related data mining methods have also been widely introduced and have become one of the important research directions in mineral prediction.

Wu-Zheng-Dao District in China is the world’s most famous mining areas subject to over a century of mineral exploration. And it hosts several world-class deposits, such as Xinming bauxite deposit, Datang bauxite deposit and Luolong bauxite deposit. Although this area still have significant potential for discovery of new deposits, mineral prediction is becoming more and more diffculity as the number of shallow deposits is diminishing.. The prediction model based on GIS is one of the effective means for comprehensive metallogenic prediction, because: (1) it can objectively and accurately evaluate the control of various prospecting factors (including magmatic rocks, strata, structures, geophysical and geochemical anomalies, etc.) On metallogenesis in the region; (2) It can effectively and comprehensively study the prospecting factors of different sources in the region, select appropriate spatial mathematical models, summarize the spatial distribution law of various prospecting factors and deposits, and predict and evaluate the metallogenic potential in the region.

Based on the rock geochemical data of different geochemical data in Wu-Zheng-Dao area, this study selects various prospecting factors, adopts evidence weight and machine learning to predict mineralization. The parameters of different algorithms and the influence of sample selection on the prediction results are summarized, and the prediction results of different models are comprehensively evaluated.Spatial analysis and evidence weight analysis are used to analyze the correlation between each prospecting factor and the spatial distribution of deposits in Wu-Zheng-Dao area, and the evidence weight method is used to predict mineralization.[1-4].

4 Specify the research gap or novelty that your study addresses.

Added this section in the conclusion section：(5)The previous ore-forming prediction work in the northern Guizhou region mainly focused on qualitative research. Applying quantitative methods to the exploration of bauxite deposits in the northern Guizhou region can provide a portable method system and quantitative model support for future ore-forming prediction work in the region. At the same time, clay layer anomalies have been overlooked in previous research and exploration work. The weight of evidence and machine learning prediction models in this article consistently indicate that clay anomalies contribute only to mineralization prediction, second only to aluminum anomalies, and are one of the most explicit prospecting indicators at the regional scale. This discovery can provide important new ideas and clues for prospecting and exploration in this area, as well as new evidence for the contribution of regional surrounding rocks to bauxite mineralization.

5 Expand the literature review to include recent and relevant studies in the field, emphasizing the gaps your research aims to fill. For instance

a. Mitigating tribological challenges in machining additively manufactured stainless steel with cryogenic-MQL hybrid technology

b. Parallel structure of crayfish optimization with arithmetic optimization for classifying the friction behaviour of Ti-6Al-4V alloy for complex machinery applications

c. A synergy of an evolutionary algorithm with slime mould algorithm through series and parallel construction for improving global optimization and conventional design problem

d. Pelton wheel bucket fault diagnosis using improved shannon entropy and expectation maximization principal component analysis

e. An adaptive feature mode decomposition based on a novel health indicator for bearing fault diagnosis

This is a great suggestion. These articles are all excellent, and we have carefully read them, made modifications and adjustments to this section, and cited references.References [7]-[11]

6 Provide more context on the specific challenges or issues addressed by your research.

Added chapter :Discussion.The evidence weighting method itself has certain limitations, including requiring the input data to follow a certain distribution pattern and the need for conditional independence between input features. To overcome these problems, many scholars have proposed improved methods for the evidence weight method, such as Agterberg's weighted evidence weight model based on logistic regression method, which uses logistic regression coefficients as weighting factors to obtain a mixed calculation model of logistic regression and evidence weight method, effectively solving the problem of maintaining conditional independence of each evidence layer in the evidence weight model; Cheng Qiuming proposed a theoretical model for enhancing evidence rights, and compared it with ordinary evidence rights methods through examples. The results showed that the enhanced evidence rights method significantly improved the practical application effect; Sun Tao et al. used the evidence weight method combined with fractal method to delineate the tungsten mineralization prospects in the southern Jiangxi region, and achieved good prediction results. The commonly used shallow machine learning methods in the field of mineralization prediction include support vector machines, random forests, and artificial neural networks. Zuo et al. applied the support vector machine algorithm to predict mineralization prospects. Research has shown that the support vector machine algorithm integrates multiple evidence variables for mineralization prospect prediction, and its results are significantly better than the evidence weight method. Du et al; Wang et al. used the semi supervised random forest method to map the mineralization prospects of the southwestern Fujian metallogenic belt, indicating that semi supervised learning can improve the effectiveness of mineralization prospect mapping; Li et al. used the random forest algorithm based on bagging technology to predict the metallogenic prospect of tungsten polymetallic deposits in the the Nanling Mountain metallogenic belt, which overcame the uncertainty caused by the discrete evidence map and the lack of data caused by the scarcity of deposits/ore occurrences. In recent years, with the maturity of machine learning in the field of mineralization prediction, coupled with the rapid growth of geological data and the continuous improvement of computer technology, a large number of experts and scholars have pushed deep learning based mineralization prediction onto the research boom. Ghezelbash used two supervised machine learning algorithms, radial basis function neural network and support vector machine with RBF kernel, to generate a data-driven porphyry copper deposit prediction model. The results showed that the radial basis function neural network had the best performance in delineating the ore-forming prospect area. Machine learning algorithms have strong ability to handle nonlinear spatial relationships and can integrate massive multi-source spatial data. There are various types of machine learning algorithms, and different algorithms exhibit inconsistent predictive performance in different regions. Therefore, in the face of the multi-source, heterogeneous, and nonlinear characteristics of geological big data in the current era, this article combines previous research and fully explores multi-source geological information. Based on this, shallow machine learning algorithms such as random forest and neural networks, as well as typical deep learning algorithms, are combined with geological big data to carry out tungsten mineralization prediction work in the study area, aiming to seek the best prediction algorithm in the study area. And achieved certain results.

7  Provide more detailed insights into the findings, explaining the practical implications of different motion states observed.

The article has been supplemented and modified to some extent

8 Consider including more visual aids (charts, graphs) to enhance the presentation of results.

Supplemented the conclusion section

9 Summarize the key findings in a concise manner, emphasizing their significance and practical implications..

The article has been supplemented and modified to some extent

10 Consider explicitly stating how your findings contribute to the existing knowledge in the field.

Supplemented the conclusion section

11 Include a section on potential future work or research directions based on the current findings.

Shortcomings and Future Work Prospects 

(1)The information related to mineralization needs to be comprehensive. Deep prediction is the focus of future mineral exploration work, and the quantitative prediction method research approach in this article is also applicable to deeper resource exploration. However, more layers of deep mineralization information need to be added to transform the planar prediction problem in this article into three-dimensional prediction; From the analysis of the prediction results in this article, it can be seen that the contribution of geophysical information reflecting deep elements to the prediction results is the worst, indicating that there are certain deficiencies in the resolution and indicative nature of the deep mineralization information layer collected in this article at the regional (local) mining area scale. 

(2)The insufficient sample size in the training set is a challenge in 

---

## [Decision Letter · Decision Letter 1]

9 May 2024

PONE-D-24-02196R1Prediction and practical application of bauxite mineralization in Wuzhengdao area, Guizhou, ChinaPLOS ONE

Dear Dr. Yang,

Thank you for submitting your manuscript to PLOS ONE. After careful consideration, we feel that it has merit but does not fully meet PLOS ONE’s publication criteria as it currently stands. Therefore, we invite you to submit a revised version of the manuscript that addresses the points raised during the review process.

**The manuscript has been evaluated by four reviewers, and their comments are available below. Please review these and make any necessary changes to the manuscript.**

We look forward to receiving your revised manuscript.

Kind regards,

Joanna Tindall

Staff Editor

PLOS ONE

Journal Requirements:

Reviewers' comments:

Reviewer's Responses to Questions

**Comments to the Author**

1. If the authors have adequately addressed your comments raised in a previous round of review and you feel that this manuscript is now acceptable for publication, you may indicate that here to bypass the “Comments to the Author” section, enter your conflict of interest statement in the “Confidential to Editor” section, and submit your "Accept" recommendation.

Reviewer #1: All comments have been addressed

Reviewer #2: All comments have been addressed

Reviewer #3: All comments have been addressed

Reviewer #4: All comments have been addressed

2. Is the manuscript technically sound, and do the data support the conclusions?

Reviewer #1: Yes

Reviewer #2: Yes

Reviewer #3: Yes

Reviewer #4: Yes

3. Has the statistical analysis been performed appropriately and rigorously? 

Reviewer #1: Yes

Reviewer #2: Yes

Reviewer #3: Yes

Reviewer #4: Yes

4. Have the authors made all data underlying the findings in their manuscript fully available?

Reviewer #1: Yes

Reviewer #2: Yes

Reviewer #3: Yes

Reviewer #4: Yes

5. Is the manuscript presented in an intelligible fashion and written in standard English?

Reviewer #1: Yes

Reviewer #2: Yes

Reviewer #3: Yes

Reviewer #4: Yes

6. Review Comments to the Author

**Reviewer #1:** As the study analyzes the factors that influence/contribute to the potential of three different ML algorithms to produce accurate results when used to explore a mineral deposit. As discussed in the conclusion, each method has a different data structure. This difference has produced different results. On the other hand, it is not possible to test the success of these methods in a field that is still under exploration (reserve and metal content information is unknown). In this case, it would not be meaningful to compare the methods.

In conclusion, although this study discusses the predictive efficiency of three different algorithms, there is no criterion (reserve and metal content information) to prove that each of them is appropriate or not. Therefore, there is no conclusion reached by the study. In other words, the appropriate methodology and/or the data set that should be used for a successful prospecting etc. are not proposed.

**Reviewer #2**: All the comments have been addressed and the suggestions have been duly incorporated within the manuscript.

**Reviewer #3: **The significance of the study is not well emphasized. This should be explained concisely.

Grammar check is required all through the manuscript. Check page 13 line 58 for correction. A word appears incorrectly written.

Page 13 line 72 should be prediction-based model ….

Authors should explain why the sensitivity of logistic regression decreased to varying degrees

**Reviewer #4: **(No Response)

7. PLOS authors have the option to publish the peer review history of their article (what does this mean?). If published, this will include your full peer review and any attached files.

Reviewer #1: **Yes: **Kerim AYDINER

Reviewer #2: No

Reviewer #3: No

Reviewer #4: No

---

## [Author Response · Author response to Decision Letter 1]

28 May 2024

Dear editor：

Thank you very much for your letter. We have learned much from two reviewers’ comments, which are fair, encouraging and constructive. After carefully studying the comments and your advice, we have made corresponding changes. The main revisions are listed below.

For reviewer one:

Reviewer #1: As the study analyzes the factors that influence/contribute to the potential of three different ML algorithms to produce accurate results when used to explore a mineral deposit. As discussed in the conclusion, each method has a different data structure. This difference has produced different results. On the other hand, it is not possible to test the success of these methods in a field that is still under exploration (reserve and metal content information is unknown). In this case, it would not be meaningful to compare the methods.

In conclusion, although this study discusses the predictive efficiency of three different algorithms, there is no criterion (reserve and metal content information) to prove that each of them is appropriate or not. Therefore, there is no conclusion reached by the study. In other words, the appropriate methodology and/or the data set that should be used for a successful prospecting etc. are not proposed

Thanks to the opinions of the reviewer, The opinions of the reviewers are very precise and directly reveal the problems existing in our discipline, which is worthy of our deep consideration and change. Mineral prediction has always been called pseudoscience. Different models, different algorithms and different people will have different results in the same area, because there are too many boundary conditions in our geological conditions, and many results are unknown.

As the reviewer said, As discus sed in the conclusion, each method has a different data structure. This difference has produced different results”,“ it is not possible to test the success of these methods in a field that is still under exploration”Indeed, what we are doing is not precise work. This is different from the fine research in the field of materials metallurgy. Our work can only screen out relatively effective methods, which increases the probability of mineral exploration. This method is suitable for regions with the same geological background to find similar types of minerals. At the same time, we have had successful examples of application in the region. Our mineral exploration results have been submitted to the Department of Natural Resources of Guizhou Province and have been recognized by the country. Here is the approval given by our government as support, hoping to receive some recognition and recognition from the reviewers.

For reviewer two:

1 The significance of the study is not well emphasized. This should be explained concisely.

Grammar check is required all through the manuscript. Check page 13 line 58 for correction. A word appears incorrectly written.Page 13 line 72 should be prediction-based model ….Authors should explain why the sensitivity of logistic regression decreased to varying degrees

We add a paragraph highlighting the research implications. 

This work can provide ideas for geological analogy and geological research in regional prospecting projects, can concentrate on summarizing complex geological phenomena into some data parameters, can provide a complete concept of metallogenesis, is conducive to the study of metallogenic environment and the distinction between metallogenic environment and non-metallogenic environment, and can formulate a reasonable and economic prediction scheme for regional prospecting; It is an indispensable link in the process of transformation from qualitative to quantitative research

We have invited professional English institutions to polish and revise the language of the article. 

Yang Wu

---

## [Editor Report · Decision Letter 2]

7 Jun 2024

Prediction and practical application of bauxite mineralization in Wuzhengdao area, Guizhou, China

PONE-D-24-02196R2

Dear Dr. Yang,

We’re pleased to inform you that your manuscript has been judged scientifically suitable for publication and will be formally accepted for publication once it meets all outstanding technical requirements.

Kind regards,

Joanna Tindall

Staff Editor

PLOS ONE
---

## [Editor Report · Acceptance letter]

24 Jun 2024

PONE-D-24-02196R2 

PLOS ONE

Dear Dr. Yang, 

I'm pleased to inform you that your manuscript has been deemed suitable for publication in PLOS ONE. Congratulations! Your manuscript is now being handed over to our production team.

Kind regards, 

on behalf of

Dr Joanna Tindall 

Staff Editor

PLOS ONE